# Structural basis for DNA 3′-end processing by human tyrosyl-DNA phosphodiesterase 1

Fiona J. Flett[1], Emilija Ruksenaite [2], Lee A. Armstrong[2], Shipra Bharati[2], Roberta Carloni[1], Elizabeth R. Morris [2], C. Logan Mackay[3], Heidrun Interthal[1] & Julia M. Richardson[2]

Tyrosyl-DNA phosphodiesterase (Tdp1) is a DNA 3′-end processing enzyme that repairs topoisomerase 1B-induced DNA damage. We use a new tool combining site-specific DNA–protein cross-linking with mass spectrometry to identify Tdp1 interactions with DNA. A conserved phenylalanine (F259) of Tdp1, required for efficient DNA processing in biochemical assays, cross-links to defined positions in DNA substrates. Crystal structures of Tdp1–DNA complexes capture the DNA repair machinery after 3′-end cleavage; these reveal how Tdp1 coordinates the 3′-phosphorylated product of nucleosidase activity and accommodates duplex DNA. A hydrophobic wedge splits the DNA ends, directing the scissile strand through a channel towards the active site. The F259 side-chain stacks against the −3 base pair, delimiting the junction of duplexed and melted DNA, and fixes the scissile strand in the channel. Our results explain why Tdp1 cleavage is non-processive and provide a molecular basis for DNA 3′-end processing by Tdp1.

---

[1] Institute of Cell Biology, School of Biological Sciences University of Edinburgh, The King's Buildings, Roger Land Building, Alexander Crum Brown Road, Edinburgh EH9 3FF, UK. [2] Institute of Quantitative Biology, Biochemistry and Biotechnology, School of Biological Sciences, University of Edinburgh, The King's Buildings, Max Born Crescent, Edinburgh EH9 3BF, UK. [3] EaStCHEM School of Chemistry University of Edinburgh, The King's Buildings, David Brewster Road, Edinburgh EH9 3FJ, UK. Fiona J. Flett and Emilija Ruksenaite contributed equally to this work. Correspondence and requests for materials should be addressed to H.I. (email: heidrun.interthal@ed.ac.uk) or to J.M.R. (email: julia.richardson@ed.ac.uk)

Normal DNA metabolism can create lesions that, if not repaired, can lead to genome instability, contributing to diseases including cancer and neuropathies[1]. Knowing how DNA is damaged and how cells deal with damaged DNA is not only important for understanding the aetiology of these diseases, but it can also be exploited for anti-cancer therapy.

Broken DNA strands, created by exogenous or endogenous DNA damage, often contain lesions that block their 3′ or 5′ ends. These must be removed before the broken strand can be repaired. The DNA repair enzyme tyrosyl-DNA phosphodiesterase 1 (Tdp1) removes 3′-blocking lesions and then protects 3′ ends from further degradation during DNA repair. Tdp1 confers a neuroprotective function in vivo[2], by facilitating single-strand break repair in neurons. A mutation in the Tdp1 active site causes the rare neurodegenerative disorder spinocerebellar ataxia with axonal neuropathy[3].

Human Tdp1 repairs DNA damage created by topoisomerase 1 (Top1)[4–7], an enzyme that relaxes superhelical tension in DNA. To enable strand rotation, Top1 generates a temporary single-strand break (SSB), by covalently attaching to the DNA 3′-end at the SSB via a phosphotyrosine bond. Normally the Top1–DNA complex is transient, but if it persists, the SSB can become a cytotoxic DNA double-strand break (DSB), for example, during DNA replication[8,9] or RNA transcription[10–12]. Camptothecin (CPT)-based anticancer therapies exploit this weakness to promote the death of rapidly replicating cancer cells. By intercalating between Top1 and the nicked DNA, they stall Top1–DNA complexes[13] and prevent DNA religation[14].

Tdp1 hydrolyses the 3′-phosphotyrosine linkage between Top1 and DNA (Fig. 1a), releasing Top1 and preventing DSBs. Thus, Tdp1 counters the action of CPT-based cancer therapies and is therefore a promising target to augment chemotherapy[15,16]. The development of drugs to inhibit Tdp1 function will be informed by understanding how Tdp1 interacts with and processes physiological DNA substrates.

Tdp1 can also process a variety of other physiological, pharmacological and synthetic phospho-DNA adducts. These include 3′-phosphoglycolates[17] (Fig. 1b), produced by exogenous oxidative DNA damage; unmodified DNA with a 3′-hydroxyl end[18] (Fig. 1c); and chain-terminating nucleoside analogues (CTNAs) (Fig. 1d), such as acyclovir, zidovudine (AZT) and cytarabine—widely used antiviral or anticancer drugs that target DNA polymerases[19]. $Tdp1^{-/-}$ cells are hypersensitive to these treatments and accumulate DNA damage, implicating Tdp1 in repair of CTNA-induced DNA damage in vivo[20]. Furthermore, Tdp1 cleaves synthetic 3′-adducts (such as artificial biotin, fluorophores and black hole quenchers), used to dissect Tdp1's activity in vitro[18,21] and to screen for Tdp1 inhibitors[16,22].

After removing a 3′-blocking lesion, Tdp1 leaves a 3′-phosphate on the DNA end, which it cannot hydrolyse further[18]. Thus, Tdp1 cleavage is non-processive. The 3′-phosphate protects the DNA end before downstream enzymes (e.g., polynucleotide kinase phosphatase) convert it to a 3′-OH, prior to repair by DNA polymerases and ligases.

X-ray crystal structures gave initial insights into the mechanism by which Tdp1 cleaves DNA–phosphotyrosine bonds[23–26]. These revealed the architecture of the active site and how it coordinated a transition-state intermediate mimic, comprising a Top1-derived peptide connected to a short single-stranded DNA via a vanadate ion[25]. Only three unpaired nucleotides were observed in a narrow, positively charged cleft extending from the Tdp1 active site. However, the DNA component of a biological Tdp1 substrate is most likely a nicked double-stranded species[27,28], as shown in Fig. 1a. How Tdp1 recognises the duplex structures of normal substrates is therefore an open question.

We use an integrated structural approach to establish how Tdp1 accommodates physiological DNA substrates. We develop a new tool to identify Tdp1–DNA interactions by combining site-specific cross-linking of protein to modified DNA with mass spectrometry. The findings are validated by two new crystal structures of Tdp1 in complex with duplex DNA, which capture the DNA repair machinery after 3′-end cleavage. These reveal how a hydrophobic loop separates the DNA strands. Coordination of the 3′-phosphorylated DNA end in the active site and stacking of a well-conserved phenylalanine against the −3 base fixes the scissile strand position and ensures that 3′-cleavage by Tdp1 is not processive. Our results explain how Tdp1 removes blocking lesions on duplex or single-stranded DNA and provide a structural basis for Tdp1's ability to protect 3′-phosphorylated DNA ends.

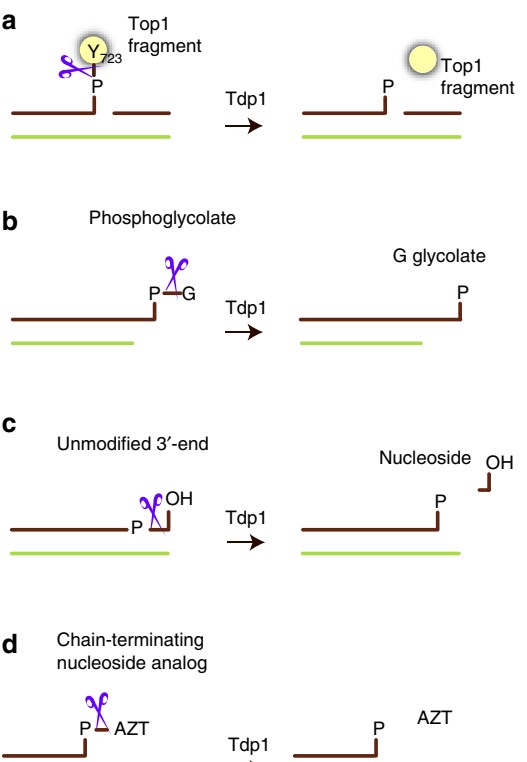

**Fig. 1** Tdp1 is a DNA 3′-end processing enzyme. Schematic representation of Tdp1 activity (represented by scissors) on biologically and medically relevant substrates. All Tdp1 reactions result in DNA with a 3′-phosphorylated end. **a** Hydrolysis of the phosphotyrosyl linkage between a proteolytic topoisomerase 1 fragment and the 3′-end of the DNA at a nick. **b** Removal of glycolate from 3′ overhangs with a phosphoglycolate (PG) adduct. **c** Tdp1 nucleosidase activity on unmodified DNA with a 3′-hydroxyl end. **d** Removal of chain-terminating nucleoside analogues (CTNAs), such as AZT (zidovudine) from a recessed 3′ end

## Results

**Site-specific protein–DNA cross-links and mass spectrometry.** We developed a new protocol (Fig. 2) using site-specific UV cross-linking of DNA to protein, combined with mass spectrometry to identify residues in human Tdp1 that are in close proximity to specific positions in substrate DNA. Tdp1-bound oligonucleotides containing the photoactivatable nucleotide 5-Iodouracil (5IdU)[29,30] and a 3′ biotin-TEG tag were covalently cross-linked to proximal aromatic amino acids by irradiation with UV at 312 nm. By changing the position of the 5IdU on the DNA,

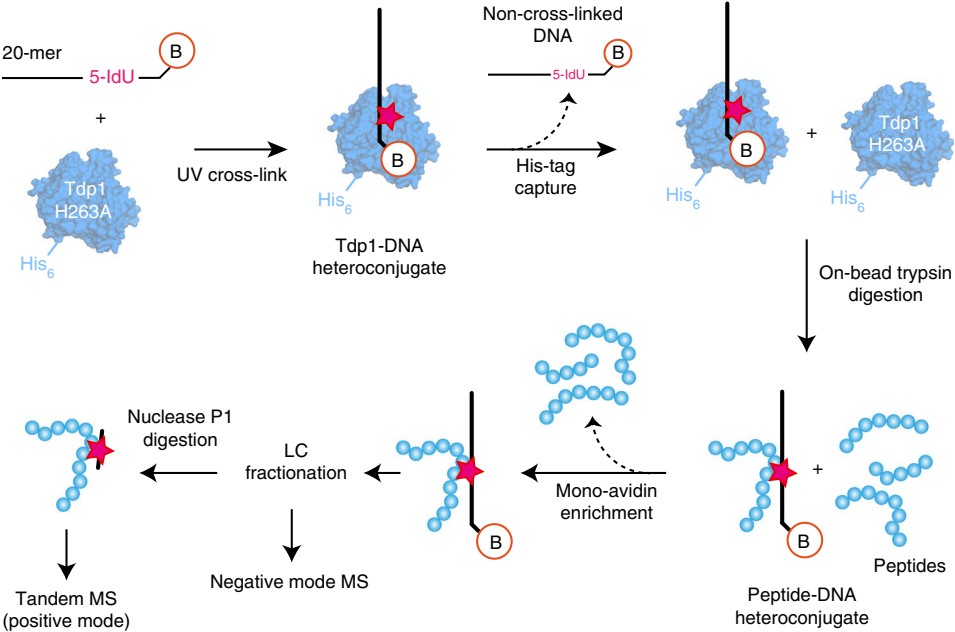

**Fig. 2** PD-XLMS method for site-specific protein–DNA cross-linking with mass spectrometry. Tdp1(Δ148) protein (blue, H263A inactive mutant) is cross-linked to DNA (black line) containing 5-Iodouracil (5IdU) and a 3′-Biotin-TEG (depicted as B in a red circle). Small blue circles represent Tdp1 amino acids (after trypsin digestion) and the red star indicates the site of the cross-link

different Tdp1–DNA cross-links were formed. In this way interactions between the known DNA positions and the protein can be identified and mapped.

The application of chemical cross-linking with mass spectrometry (XLMS) to mapping protein–DNA interactions is considerably less well established[31,32] than techniques for analysing protein–protein or protein–RNA interactions[33,34]. This is mainly due to challenges in achieving specific protein–DNA cross-links without background protein–protein cross-linking and with low yields of cross-linked heteroconjugates[35]. Our PD-XLMS protocol (Fig. 2) incorporates new enrichment and purification steps that overcome these issues.

First, cross-linked Tdp1–DNA heteroconjugates were enriched by capturing Tdp1 via its hexa-histidine tag on magnetic beads, concomitantly removing non-cross-linked DNA. Tdp1–DNA heteroconjugates were then digested with trypsin, releasing peptide–DNA heteroconjugates from the beads. Heteroconjugates were enriched by binding to monomeric avidin via the 3′ biotin-TEG tag, at the same time removing non-conjugated peptides. Subsequently, samples were characterised by our previously described mass spectrometry protocol[36]: analysis by LC/MS in negative mode identified Tdp1 peptide–DNA heteroconjugates. Then, after nuclease P1 digestion, tandem MS enabled the Tdp1 amino acid cross-linked to DNA to be identified.

**Tdp1 cross-links to modified nucleobases via F259.** To establish site-specific cross-links, 20-mer single-stranded DNA oligonucleotides containing 5IdU at either the −2 or the −3 position, or unmodified as a control (Fig. 3a and Table 1), were incubated with truncated Tdp1 (amino acids 149–608), hereafter referred to as Tdp1(Δ148). Use of the catalytic mutant H263A Tdp1(Δ148) ensured that the 3′ biotin-TEG, a non-physiological substrate of Tdp1[18], was not removed by the enzyme. Following UV irradiation, reactions were analysed by SDS-PAGE and visualised by phosphorimaging (Fig. 3b, upper panel, and Supplementary Fig. 1A) and SimplyBlue™ staining (Fig. 3b, lower panel, and Supplementary Fig. 1B). $^{32}$P-labelled species migrating more slowly than substrate DNA were predicted to be hetero-conjugates in which either the −2 or the −3 5IdU oligonucleotide was specifically cross-linked to Tdp1(Δ148) (lanes 2 and 3, Fig. 3b, upper panel). These species were absent in the control reaction performed with the unmodified oligonucleotide (lane 1, Fig. 3b, upper panel). The species also migrated more slowly than free, non-cross-linked Tdp1(Δ148) protein on SimplyBlue™-stained SDS-PAGE (Fig. 3b, lower panel), consistent with formation of heteroconjugates. Thus, Tdp1(Δ148) was specifically cross-linked to the modified 20-mer oligonucleotides.

Next, we used mass spectrometry to confirm the generation of Tdp1(Δ148)–DNA heteroconjugates and to identify which Tdp1 peptides were cross-linked to the modified −2 and −3 nucleobases. Both the −3 and −2 5IdU cross-linked samples contained UV-absorbing species, eluting between 15 and 17 min from a reverse-phase column (Fig. 3c), that were consistent with peptide–DNA heteroconjugates. The corresponding masses of the most abundant potential heteroconjugates in this time interval were 7927.11 Da and 7912.11 Da in the −3 and −2 cross-linked samples, respectively (Fig. 3d). Importantly, these species were absent in the control sample. From these data, we calculated the mass of the peptide cross-linked, by subtracting the mass of each oligonucleotide (−3, 6882.19 Da and −2, 6867.19 Da) minus HI (127.91 Da), which is lost during cross-linking. The mass difference in both cases corresponds to the mass (1173 Da) of the Tdp1(Δ148) tryptic peptide sequence L$_{255}$DIAFGTHATK$_{265}$.

To sequence the cross-linked peptide and to identify exactly which Tdp1(Δ148) amino acids were cross-linked to the 5IdU-modified nucleotides, Tdp1(Δ148) peptide–DNA heteroconjugates were digested with nuclease P1 to reduce the DNA moiety to a single nucleotide monophosphate. Using high-resolution tandem mass spectrometry, these digested species were then detected (Fig. 3e) and sequenced (Fig. 3f). A unique [M+2H]$^{2+}$ ion with $m/z$ 740.33 was detected in both the −3 and −2 cross-linked samples, which was absent from the control (Fig. 3e). This ion corresponds to the mass of a heteroconjugate composed of the Tdp1(Δ148) peptide L$_{255}$DIAFGTHATK$_{265}$ cross-linked to deoxyuracil monophosphate (dUMP). Collision-induced

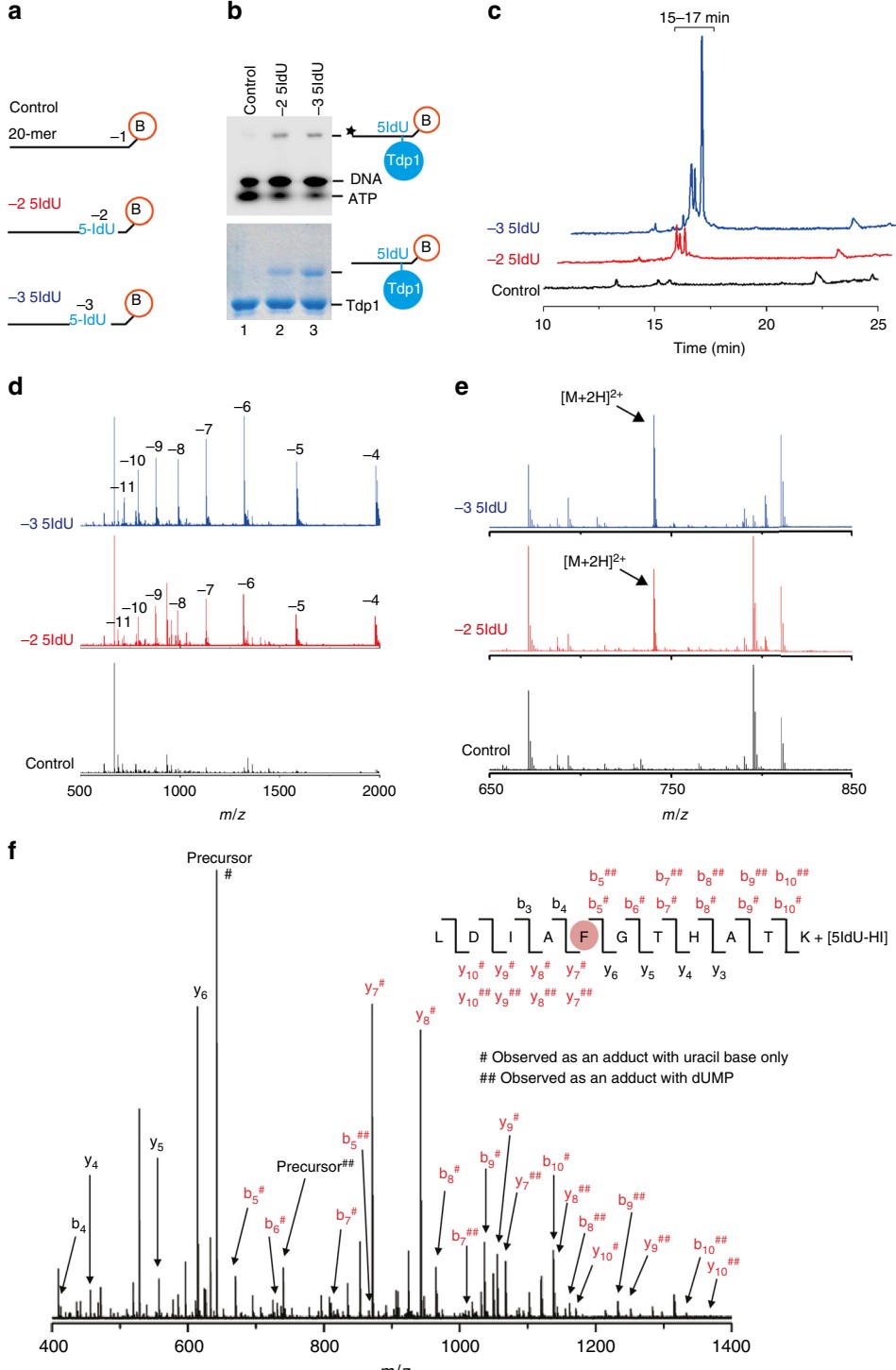

**Fig. 3** Tdp1(Δ148) cross-links via F259 to −2 and −3 modified nucleobases. **a** Schematic of the single-stranded DNA 20-mers used in cross-linking experiments. Modified oligonucleotides have 5-Iodouracil (5IdU) at the −2 or −3 position. **b** Eight per cent SDS-PAGE of DNA oligonucleotides, containing a $^{32}$P-label (star), cross-linked to catalytically inactive Tdp1(Δ148) H263A, visualised by phosphorimaging (upper) and SimplyBlue[TM] staining (lower). **c** LC elution chromatograms (at 260 nm) of samples cross-linked to the −3 5IdU (blue), −2 5IdU (red) and unmodified control (black) oligonucleotides. **d** Negative-mode mass spectra showing the charge-state distribution of cross-linked Tdp1(Δ148) peptide–DNA heteroconjugates (cross-linked to the −3 5IdU (blue), −2 5IdU (red) and unmodified control (black) oligonucleotides) eluting from the LC between 15 and 17 min after injection. **e** Positive-mode mass spectra of −3 5IdU (blue), −2 5IdU (red) and control (black) cross-linked, trypsin and nuclease digested samples. Unique [M+2H]$^{2+}$ ions (marked by arrows) are observed at $m/z$ 740.33269 with both 5IdU cross-linked samples. **f** Collision-induced dissociation fragmentation (CID) mass spectrum of the [M+2H]$^{2+}$ ion at $m/z$ 740.33 at 20 V (for the −2 cross-link sample) and the sequence of the cross-linked Tdp1 peptide. The position of the DNA cross-link (F, shaded in red) is determined by the presence of modified fragment ions (annotated in red) containing a single deoxyuracil monophosphate (dUMP, annotated with ##) or a uracil base (annotated by #), which arises from fragmentation of the glycosidic ribose–base bond during CID. Identified peptide *b* and *y* fragment ions are annotated in black

**Table 1 Sequences of the oligonucleotides used in the cross-linking, fluorescence and crystallisation experiments**

| Name | Sequence (5′–3′) | Length (nt) |
|---|---|---|
| Tdp1(Δ148)–DNA cross-linking | | |
| Control | $^{32}$P-GTAGAGGATCTAAAAGACTT-Biotin-TEG | 20 |
| 5IdU(−2) | $^{32}$P-GTAGAGGATCTAAAAGAC(5IdU)T-Biotin-TEG | 20 |
| 5IdU(−3) | $^{32}$P-GTAGAGGATCTAAAAGA(5IdU)TT-Biotin-TEG | 20 |
| Fluorescence quenching assays | | |
| HEI-40 | 56-FAM/AGA GGA TCT AAA AGA CTT/3BHQ | 18 |
| HEI-50 | 56-FAM/AAG TCT TTT AGA TCC CTC CGG ATC TAA AAG ACT T/3BHQ | 34 (15 nt hairpin) |
| Crystallisation | | |
| −2G 12-mer | CACTGCGCAGTG | 12 |
| −2T 12-mer | CAATGCGCATTG | 12 |
| 14-mer marker | CACTGTCGACAGTG | 14 |
| 16-mer marker | CACTGTCTAGACAGTG | 16 |
| Gel analysis of crystals | | |
| 12-mer substrate | CACTGCGCAGTG | 12 |
| 11-mer marker | CACTGCGCAGT | 11 |
| 10-mer marker | ACTGCGCAGT | 10 |
| 9-mer marker | CTGCGCAGT | 9 |
| 8-mer marker | TGCGCAGT | 8 |
| 7-mer marker | GCGCAGT | 7 |

Modifications are $^{32}$P—5′ radioactive phosphate; Biotin-TEG—a 3′ Biotin adduct attached via a triethyleneglycol linker; 56-FAM—5(6)-carboxyfluorescein; 3BHQ—the Black Hole Quencher

fragmentation of the heteroconjugates (Fig. 3f) revealed that for the −2 cross-linked sample, the dUMP was attached to Tdp1 (Δ148) via F259. Further confirming this cross-link, in the MS spectrum shown in Fig. 3e, we also detected and fragmented the [M+3H]$^{3+}$ ion ($m/z$ 493.89) of this cross-link. Additionally, we detected the [M+2H]$^{2+}$ and [M+3H]$^{3+}$ ions ($m/z$ 892.36 and $m/z$ 595.24, respectively) of a heteroconjugate between F259 and a nucleotide dimer of dUMP and the adjacent dTMP, following incomplete digestion of the DNA. The F259 residue also cross-links to substrates containing 5IdU at the −3 position in an identical fashion (Supplementary Fig. 2).

**Efficient DNA processing requires F, W or Y at position 259.** To establish if F259 is important for Tdp1(Δ148) activity in vitro, we substituted F259 with alanine, tryptophan or tyrosine and compared the ability of wild-type (WT) Tdp1(Δ148) and the F259A, F259W and F259Y mutants to cleave a 3′ quencher in fluorescence-based assays. Two DNA substrates were used: an 18 nt single-stranded DNA (Fig. 4a) and a double-stranded DNA hairpin containing 15 base pairs (Fig. 4b and Supplementary Fig. 3). Each substrate had a 5′ fluorophore and a 3′ quencher that ablates fluorescence (Table 1). As the 3′ quencher is removed by WT Tdp1(Δ148), fluorescence increases and reaches a steady-state level (Fig. 4c, d). Direct comparison of fluorescence intensity as a function of time with two different substrates is not possible due to the different environments of the fluorophores in the substrates. Substitution of the active site histidine H263 with alanine prevents Tdp1(Δ148) cleavage of the 3′ quencher and provides a negative control.

Reactions using the F259A Tdp1(Δ148) mutant were more than fivefold slower than those using WT Tdp1(Δ148), for both substrates. Using the single-stranded substrate at 25 °C, the steady-state fluorescence was achieved after 8 min with WT Tdp1 (Δ148) and after 40 min with the F259A mutant (Fig. 4c). By contrast, the reactions using the double-stranded DNA hairpin substrate reached steady state after 35 min with WT Tdp1(Δ148), but reached only 75% of the steady-state maximum fluorescence after 90 min with the F259A mutant (Fig. 4d). We conclude that F259 is important for efficient cleavage of both single- and double-stranded DNA substrates.

To assess whether the aromaticity or the hydrophobicity of F259 was crucial for efficient processing, we also compared the cleavage activity of WT Tdp1(Δ148) with the F259W and F259Y mutants. These mutants incorporate a similarly aromatic but less hydrophobic amino acid at position 259. The activity of the F259W Tdp1(Δ148) mutant was similar to WT enzyme on both substrates (Fig. 4c, d). However, reactions using the F259Y Tdp1 (Δ148) mutant were 1.5 times faster than WT reactions, reaching steady state after 5 min with the single-stranded substrate (Fig. 4c) and after 23 min with the double-stranded DNA hairpin substrate (Fig. 4d). Thus, we conclude that the presence of an aromatic amino acid at position 259 is more important than the hydrophobicity of the ring for efficient processing of both single- and double-stranded DNA substrates.

**Crystallisation of Tdp1(Δ148) complexes with duplex DNA.** To establish how Tdp1 recognises duplex DNA structures, we determined two X-ray crystal structures of Tdp1(Δ148) bound to duplex DNA. Tdp1(Δ148):DNA complexes were assembled by mixing the protein in a 1:1 molar ratio with 12-mer duplex DNA (Fig. 5a and Supplementary Fig. 4). Two different DNA sequences were designed, containing either guanine or thymine at the −2 position, and cytosine or adenine at the −11 position, but being otherwise identical (Fig. 5b and Table 1). The Tdp1(Δ148) complexes formed with these oligonucleotides are referred to as −2G or −2T, respectively. Each DNA sequence was self-complementary, so that annealing created a duplex with identical ends and twofold symmetry. Thus, regardless of which DNA duplex orientation was captured by the Tdp1 active site, identical complexes would be produced.

Crystals of Tdp1(Δ148) in complex with either duplex DNA grew in 4 days in the $P2_12_12$ space group. The −2G Tdp1(Δ148): DNA complex diffracted X-rays to 2.04 Å, whereas the −2T complex diffracted to 3.18 Å. The X-ray diffraction and refinement statistics are shown in Table 2. Each crystallographic asymmetric unit contains two Tdp1(Δ148) molecules: one has captured a DNA duplex, whereas the other has not. The DNA-bound and the DNA-free Tdp1(Δ148) molecules have similar structures and can be superimposed with a r.m.s.d. of 0.33 Å over 432 Cα atoms (Supplementary Fig. 5). Each DNA duplex

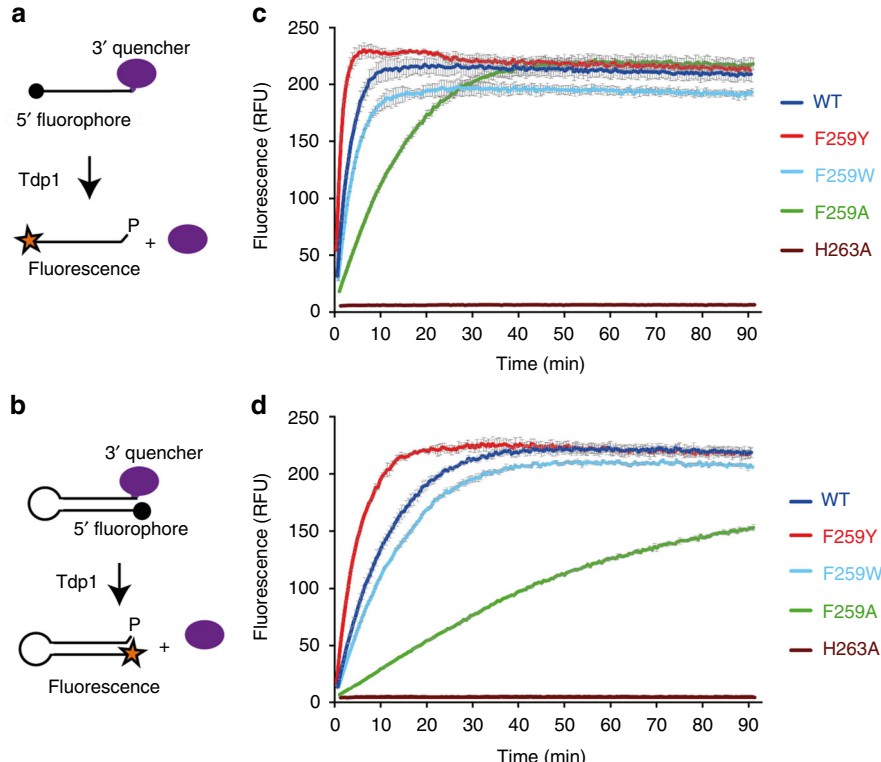

**Fig. 4** Tdp1(Δ148) activity assays. **a** Schematic of the activity assay using a single-stranded fluorescence-quenched DNA substrate. Tdp1(Δ148) cleavage of the 3′ quencher (purple ellipse) increases fluorescence of the 5′ fluorophore (black dot). **b** Schematic of the activity assay using a fluorescence-quenched hairpin substrate. **c** Plots of fluorescence intensity, in relative fluorescence units (RFU), against time in assays using the single-stranded DNA substrate and WT, F259A, F259Y, F259W or H263A Tdp1(Δ148). **d** Plots of fluorescence intensity against time for assays using the hairpin DNA substrate and WT, F259A, F259Y, F259W or H263A Tdp1(Δ148). Error bars shown in grey represent the standard deviation calculated from three experiments

connects two symmetry-related Tdp1(Δ148) molecules (Fig. 5c and Supplementary Fig. 6) and the interactions of each half of the duplex with Tdp1(Δ148) are equivalent. Therefore, we consider henceforth only one half of the 12-mer DNA duplex. We refer to the 3′ half, with the sequence 5′-GCA(G/T)TG-3′, as the scissile strand and to its complement as the complementary strand.

**Crystals contain the Tdp1 nucleosidase reaction product.** The 2.04 Å resolution crystal structure of the Tdp1(Δ148) complex with −2G duplex DNA is shown in Fig. 5c. Tdp1(Δ148) folds into two α–β–α domains, as previously described[23,24]. The active site, which comprises three pairs of conserved histidines, lysines and asparagines (H263, H493, K265, K495, N283 and N516), coordinates the 3′ end of the scissile DNA strand (Fig. 5d and Supplementary Fig. 7). Although the DNA substrate contained a 3′ guanine (0G), the electron density clearly indicates the 3′ nucleotide is phosphorylated thymine. The phosphate oxygens are within hydrogen bonding distance of H263 Hε (2.8 Å), H493 Hε (2.8 Å), K265 NHζ (3.2 Å), K495 NHζ (2.5 Å), N516 $NH_2$ (2.9 Å) and N283 $NH_2$ (3.1 Å) (Fig. 5d). The 3′-phosphate adopts tetrahedral geometry, in contrast to the trigonal bi-pyramidal coordination of the vanadate in the quaternary complex that mimics the transition state[25] (Supplementary Fig. 8). Based on this result, we hypothesised that, during preparation and crystallisation of the nucleoprotein complex, Tdp1(Δ148) removed the terminal 3′ 0G nucleoside, leaving a 3′-phosphate on the terminal −1T nucleotide. This represents a Tdp1 product complex.

To confirm Tdp1 nucleosidase activity on the 12-mer, we incubated a $^{32}$P-labelled 12-mer DNA duplex with Tdp1(Δ148)

(Fig. 6a, b, lane 1). This generated a 3′-phosphorylated 11-mer (Fig. 6b, lane 2), the expected product of Tdp1 nucleosidase activity. When separated by denaturing PAGE (Fig. 6b and Supplementary Fig. 9), the 3′-phosphorylated 11-mer (lane 2) migrates similarly to the 10-mer marker (lane 5). In contrast, adding Tdp1(Δ148) to an unlabelled 12-mer DNA duplex and subsequently $^{32}$P-labelling, the nucleosidase product using polynucleotide kinase (PNK) generates an 11-mer with a 3′OH (Fig. 6c, b, lane 3). This is because PNK removes the 3′-phosphate[37] created by Tdp1(Δ148) nucleosidase activity as well as adding $^{32}$P at the DNA 5′-ends.

The three nucleotides at the 5′ end of the complementary strand (C, A and C) were not visible in the crystal structure of the −2G Tdp1(Δ148):DNA complex. Therefore, to establish the length of the DNA contained within the crystals, we washed a single crystal in mother liquor and dissolved it in water (Fig. 6d). Following incorporation of $^{32}$P with PNK, the DNA species were separated by denaturing PAGE and the bands compared with the products of the Tdp1(Δ148) reactions described above. The dissolved crystals contained four distinct DNA species (Fig. 6b, lane 9). The main species (labelled w) comprised 60% of the mixture and migrated similarly to the 3′OH 11-mer DNA marker (lane 4), consistent with removal of the 3′ nucleoside by Tdp1 (Δ148) during ~80 days incubation in crystallo, prior to treatment with PNK (Fig. 6b, lane 3). The next most abundant DNA species in the crystal (labelled x) comprised 27% of the sample, and migrated similarly to both the 3′-phosphorylated 11-mer (lane 2) and the 10-mer marker with a 3′OH (lane 5). This species may have arisen from removal of the 3′ nucleoside by Tdp1(Δ148), but with retention of the 3′P following PNK treatment, due to reduced 3′-phosphatase activity in suboptimal

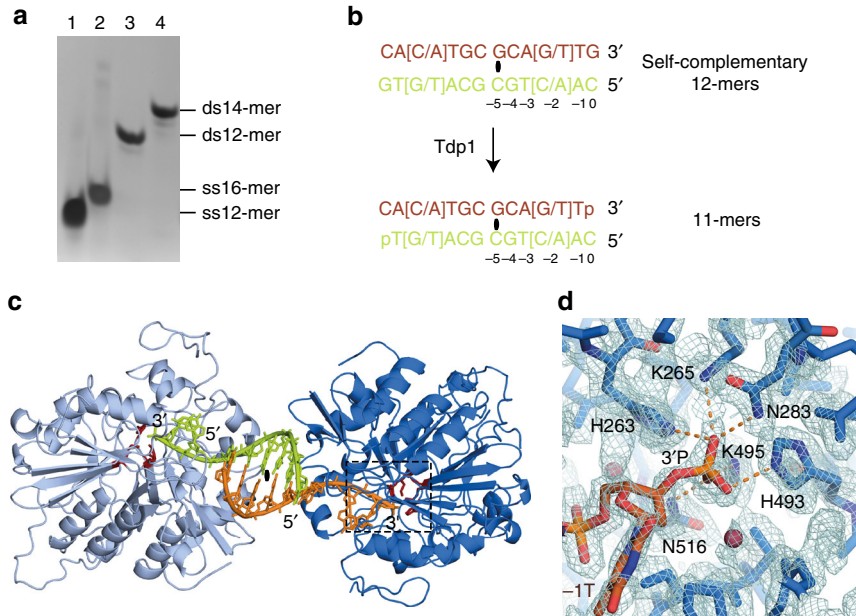

**Fig. 5** Crystallisation and X-ray structure of Tdp1(Δ148) in complex with dsDNA. **a** 20% (w/v) native polyacrylamide gel of DNA oligonucleotides stained with methylene blue. Lanes 1 and 2: single-stranded (ss) 12-mer and 16-mer, respectively; lanes 3 and 4, double-stranded (ds), self-complementary 12-mer and 14-mer, respectively. **b** Sequences of the self-complementary 12-mers used in crystallisation, containing either a G:C or a T:A base pair at the −2 position. The black ellipse indicates the centre of the palindrome. **c** Ribbon representation of the crystal structure showing two symmetry-related Tdp1 (Δ148) molecules bound to one DNA duplex. The black ellipse indicates crystallographic twofold rotational symmetry. **d** Close-up view of the Tdp1(Δ148) active site with catalytic residues labelled. Hydrogen bonds to the 3′-phosphorylated thymine (−1T) are shown as orange dashed lines and the water molecule as a red sphere. The $2F_o-F_c$ electron density map is shown as a blue mesh contoured at 1.6σ

## Table 2 Data collection and refinement statistics (molecular replacement)

|  | Tdp1(Δ148) in complex with −2G DNA duplex[a] | Tdp1(Δ148) in complex with −2T DNA duplex[a] |
|---|---|---|
| *Data collection* | | |
| Space group | $P2_12_12$ | $P2_12_12$ |
| Cell dimensions | | |
| *a, b, c* (Å) | 108.01, 195.31, 50.83 | 109.2, 198.19, 51.32 |
| *α, β, γ* (°) | 90, 90, 90 | 90, 90, 90 |
| Resolution (Å) | 52.05–2.04 | 99.1–3.18 |
|  | (2.09–2.04)[b] | (3.56–3.18–2.04) |
| $R_{sym}$ | 0.067 (1.304) | 0.315 (1.518) |
| $I/\sigma I$ | 15.3 (1.3) | 4.7 (1.4) |
| Completeness (%) | 99.2 (97.7) | 100.0 (100.0) |
| Redundancy | 9.6 (7.1) | 8.4 (8.4) |
| *Refinement* | | |
| Resolution (Å) | 52.05–2.04 | 99.1–3.18 |
| No. reflections | 69045 | 19558 |
| $R_{work}/R_{free}$ | 0.232/0.272 | 0.250/0.306 |
| No. atoms | | |
| Protein | 6894 | 7062 |
| Nucleic acid | 169 | 199 |
| Water | 57 | 0 |
| *B-factors* | | |
| Protein | 56.6 | 74.1 |
| Nucleic acid | 67.6 | 94.7 |
| Water | 62.7 | – |
| R.m.s. deviations | | |
| Bond lengths (Å) | 0.010 | 0.090 |
| Bond angles (°) | 1.26 | 1.37 |

[a] One crystal per structure
[b] Values in parentheses are for highest-resolution shell

conditions. Minor DNA species, labelled y and z (totalling 13%), migrated to the same extent as the 9-mer and 8-mer DNA markers, respectively. These may have resulted from minor shorter species present in the substrate (lane 1) or from nucleolytic degradation during prolonged incubation in the crystal. Taken together, these results support our hypothesis that the DNA contained in the crystal is the product of Tdp1(Δ148) nucleosidase activity and is longer than the 8 nt for which density is observed in the crystal structure. We conclude that this new Tdp1(Δ148):DNA co-crystal structure is a novel snapshot of the Tdp1 reaction product.

**Tdp1(Δ148) accommodates duplex DNA**. The normal in vivo DNA substrate of Tdp1 is likely a nicked duplex (Fig. 1a). But only three nucleotides of single-stranded DNA were seen in previous crystal structures of Tdp1(Δ148) transition-state inter-mediate complexes (Fig. 7a). By contrast, our new co-crystal structures reveal how Tdp1(Δ148) interacts with double-stranded DNA (Fig. 7b, c). Both strands are recognised by the charge complementarity of the protein surface to the DNA backbone atoms. Bases −5G, −4C and −3A of the scissile strand pair with the complementary strand. However, Tdp1(Δ148) separates the strands and prevents the pairing of the DNA bases at the 0, −1 and −2 positions. The 3′-end of the scissile strand is routed into a narrow, positively charged cleft (blue surface) towards the active site (Fig. 7b, c).

In the −2G Tdp1(Δ148):DNA structure, eight complementary strand nucleotides were seen (Fig. 7b). However, nine comple-mentary strand nucleotides, as well as the phosphate backbone and ribose atoms of −1A, are clearly defined by electron density in the lower resolution −2T Tdp1(Δ148):DNA structure (Fig. 7c). As the position of the −1A base in the −2T Tdp1(Δ148):DNA structure was ambiguous, we modelled this nucleotide as abasic.

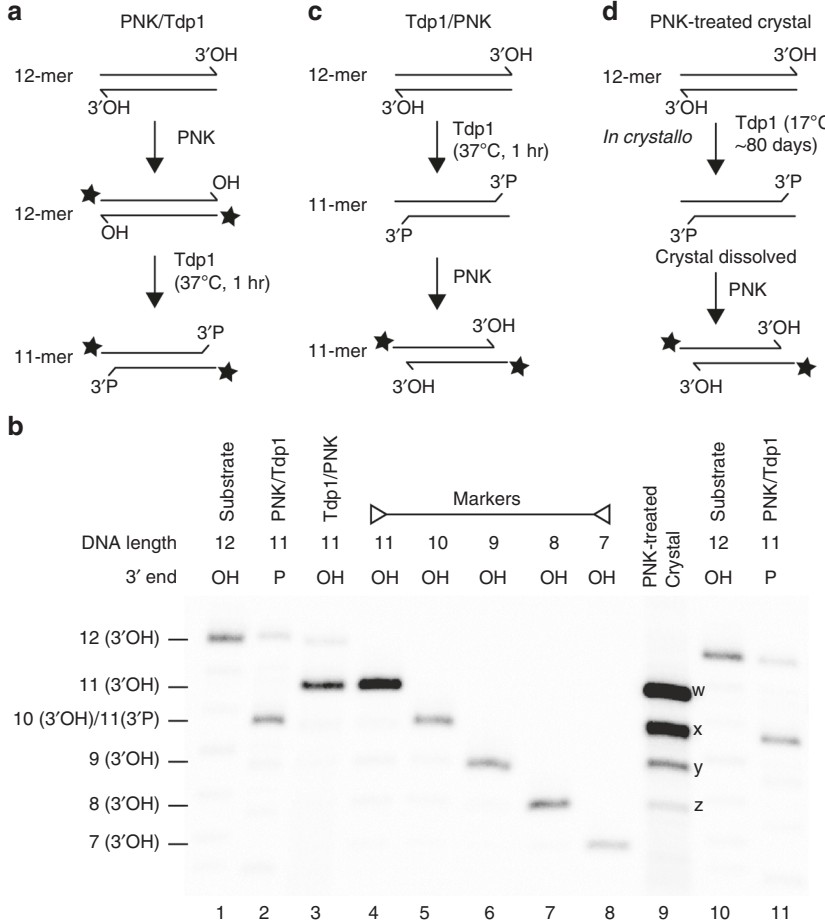

**Fig. 6** Tdp1(Δ148) nucleosidase activity and analysis of the DNA content of Tdp1(Δ148):DNA crystals. Reaction schemes depicting **a** $^{32}$P-labelling of a 12-mer DNA duplex with PNK, followed by removal of the 3′-nucleoside by Tdp1(Δ148) (star denotes $^{32}$P). **b** Denaturing PAGE of the products of the reactions shown in A (lane 2) and B (lane 3). The four distinct species in the PNK-treated dissolved crystals (lane 9) are labelled from w to z. Lanes 4–8 contain a $^{32}$P-labelled marker of length 11 nts to 7 nts, respectively. **c** 3′-nucleoside removal by Tdp1(Δ148), followed by 5′-$^{32}$P-labelling with PNK, which also removes the 3′-phosphate created by Tdp1 cleavage. **d** Proposed reaction of Tdp1(Δ148) in the crystals, followed by $^{32}$P-labelling of dissolved crystals with PNK

The 5′ 0C nucleotide was not visible in either structure and we presume it is present but disordered. This conclusion is consistent with our analysis of dissolved crystals (Fig. 6b), which shows the loss of the 3′ nucleoside (0G) only.

**Route of the complementary DNA strand**. Salt bridges and charge–charge interactions between the complementary DNA strand and Tdp1(Δ148) stabilise the co-crystal structure. The complementary strand backbone atoms follow a track of positive charges on the Tdp1(Δ148) surface (Fig. 7d) created by the side-chain amino or guanidinium groups of K231, R232, R361 and K527. The K232 amino group is in close proximity to the O3 bridging phosphate oxygen (3.1 Å) and the ribose ring oxygen (3.2 Å) of −2A; while the NHε of R361 can form a salt bridge with the non-bridging phosphate oxygen of −5C (3.6 Å). The backbone amide of K527 hydrogen bonds with the non-bridging phosphate oxygen of −6G (2.9 Å, Fig. 7e) and the $NH_2\zeta$ is 4.2 Å from the non-bridging phosphate oxygen of −5C. Consistent with this structural role, K527 is highly conserved in Tdp1 sequences (Fig. 7f), being substituted by R in *Saccharomyces cerevisiae*, which could have a similar stabilising function in that species.

K527 is part of a surface-exposed β-turn between the anti-parallel β-strands β15 and β16, which spans amino acids 527–530 (Fig. 7e). In the duplex DNA-bound Tdp1(Δ148) structures,

K527 and N528 adopt different conformations compared to previous structures containing single-stranded DNA (superimposed in Fig. 7e, PDB ID: 1NOP). This conformational change exposes a more electropositive surface along the DNA-binding cleft (Fig. 7b, c), enabling stabilising interactions between Tdp1 and the complementary strand to form. Lower B-factors of the β-turn residues, in comparison to the ssDNA bound structure (indicated by colour in Fig. 7e), reflect an ordering of the β-turn upon duplex DNA binding. Together, these structural differences are consistent with the β-turn strengthening the binding of Tdp1 to duplex DNA.

**A hydrophobic wedge separates the DNA duplex**. Tdp1(Δ148) physically blocks the pairing of the DNA bases at the 0, −1 and −2 positions using a hydrophobic loop formed by residues 257–261 (Fig. 8a). This loop precedes the active site histidine, H263, and its hydrophobic character is conserved from yeast to man (Supplementary Fig. 10). The aromatic ring of F259 intercalates between the −2 and −3 scissile strand bases, forming a π–π stack with the −3 base and demarcating the junction of melted and duplex DNA. Consistent with this structural role, F259 is strictly conserved in other Tdp1 sequences (Supplementary Fig. 10) apart from *Schizosaccharomyces pombe* Tdp1, which has W at the equivalent position. Our in vitro cleavage assays (Fig. 4) show that

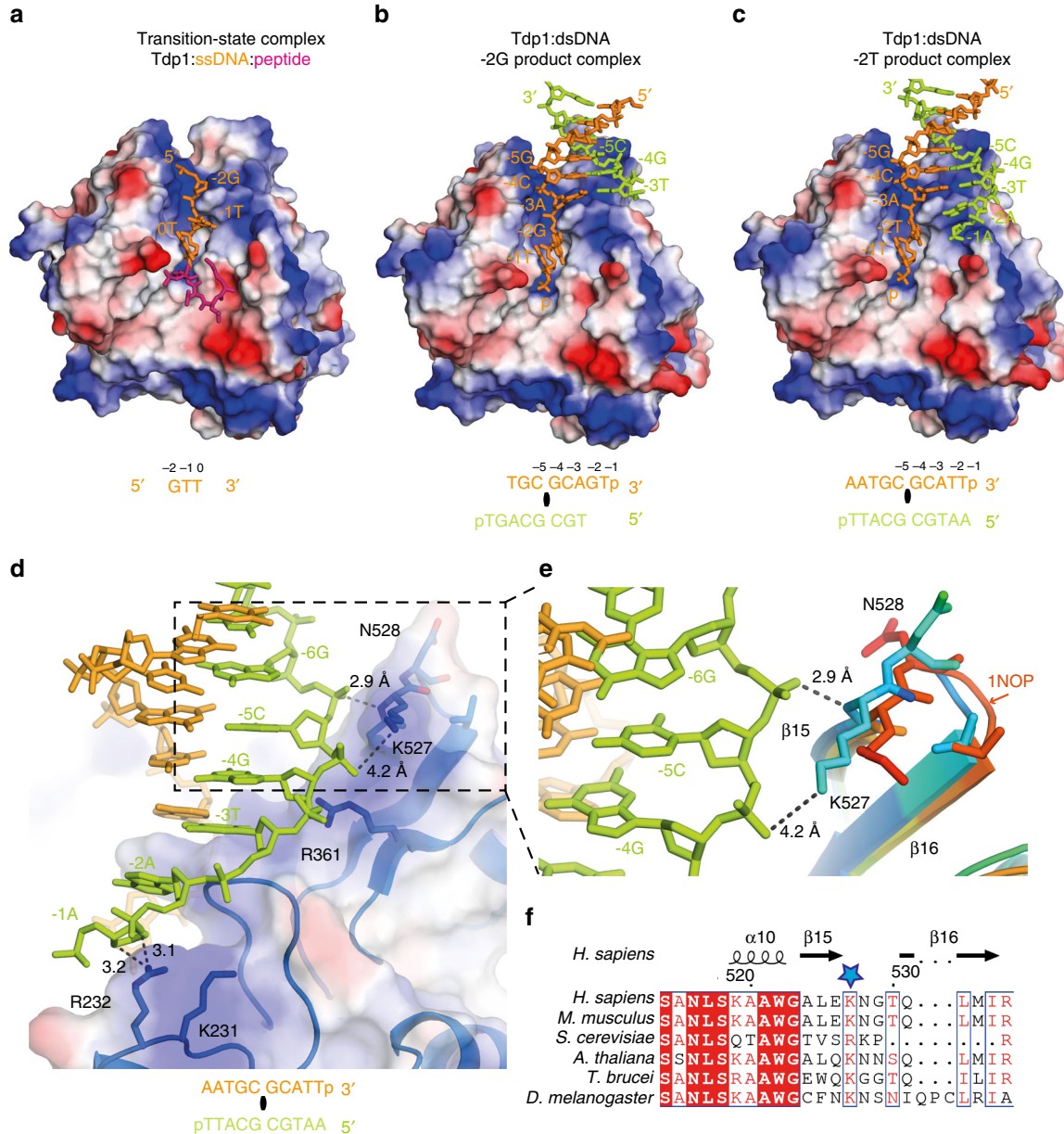

**Fig. 7** Crystal structures of Tdp1(Δ148)–DNA. **a** Quaternary transition-state complex of Tdp1(Δ148), ssDNA (orange), a Top1-derived peptide (pink) and vanadate (PDB ID: 1NOP). **b** Tdp1(Δ148) in complex with the −2G DNA duplex (PDB ID: 5NW9). **c** Tdp1(Δ148) in complex with the −2T DNA duplex (PDB ID: 5NWA). For **a**–**c**, Tdp1(Δ148) is displayed as an electrostatic surface (blue indicates positive charge and red negative charge). The sequence of the DNA in each structure that is clearly defined by the electron density is shown below. The scissile strand is orange and the complementary strand is green. **d** Tdp1(Δ148) contacts with the complementary strand in the −2T Tdp1(Δ148) DNA complex. Tdp1(Δ148) is shown as an electrostatic surface with underlying structure. The side chains of N528, K527, R361, K231 and R232 are shown as sticks, and interactions with the complementary DNA strand as dotted lines (distances in Ångstrom). **e** Close-up view of complementary strand interactions with β−turn residues K527 and N528. The structure of Tdp1 (Δ148) bound to ssDNA (PDB ID: 1NOP) is superimposed. Loop residues between β15 and β16 are rainbow coloured according to B-factors, with blue indicating the minimum and red the maximum B-factor. **f** Alignments of Tdp1 sequences from diverse species, with K527 (blue star) marked and the secondary structure elements shown above the alignment

Tdp1 with W at position 259 has similar cleavage activity to WT enzyme and thus likely performs a similar π–π stacking role to F259.

Our crystal structures support the results of the in-solution cross-linking mass spectrometry experiments, which identified short-range site-specific interactions between F259 and modified nucleobases at the −2 and −3 positions (Fig. 3). The role of F259 is also consistent with the results of our in vitro biochemical assays, which show that an aromatic amino acid side chain at

position 259 is required for efficient DNA 3′-end processing (Fig. 4) of both single- and double-stranded substrates. We envisage that, for cleavage of the single-stranded DNA substrate, the key role of the aromatic amino acid at 259 is to intercalate between nucleobases −2 and −3 and to position to scissile strand in the active site for cleavage. However, cleavage of a double-stranded substrate involves additional reshaping of the substrate, including separation of the two strands of the DNA duplex by the hydrophobic loop and

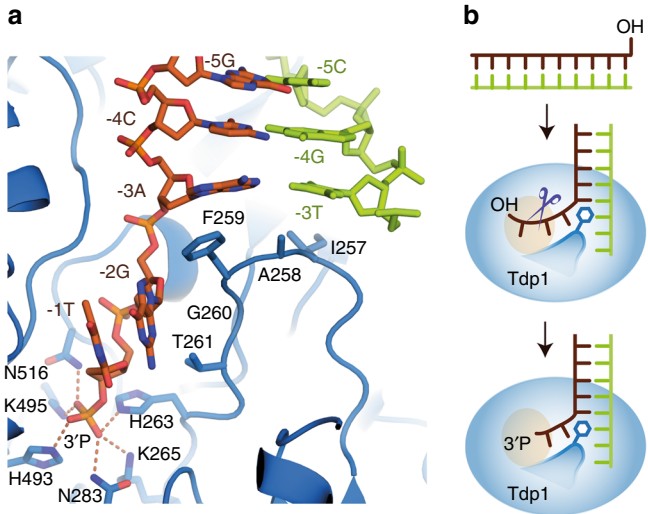

**Fig. 8** A hydrophobic wedge containing F259 disrupts base pairing and separates the DNA strands. **a** Close-up view of the hydrophobic wedge residues (I257–T261) with the scissile (orange) and complementary (green) DNA strands, shown as sticks. **b** Schematic representation of Tdp1 (Δ148) 3′ DNA processing and the role of the hydrophobic wedge

stabilisation of the melted conformation by π–π stacking with the aromatic ring of F259.

## Discussion

The results presented here for Tdp1(Δ148) interactions with DNA demonstrate the power of PD-XLMS for dissecting protein–DNA interactions in a biological system, especially in combination with high-resolution crystal structures and biochemical activity assays. Incorporating a single photoactivatable nucleotide at a defined position within the DNA substrate allows site-specific cross-linking to protein. This eliminates any ambiguity in the cross-link site within the DNA and enables identification of the directly attached amino acid by mass spectrometry. By moving the position of the modified base along the DNA, many different non-covalent protein–DNA interactions can be stabilised and then characterised. In this way, sites of protein–DNA interactions can be mapped and a model of DNA-binding developed.

Our method of PD-XLMS also offers other advantages. Performing cross-linking with low-energy UV radiation minimises DNA photo-damage and maintains the native 3D structure of the DNA–protein complex. Unique to our approach, heteroconjugates were selectively enriched prior to mass spectrometry by avidin capture of biotin-tagged DNA. Unlike titanium dioxide-based enrichment[32,38], our method is not limited to small nucleic acid moieties and is not hindered by simultaneous enrichment of charged peptides. Our recently developed mass spectrometry methods[36] allowed us to observe heteroconjugates in both negative and positive ion mode. Here, mass spectrometry was performed on an FT-ICR MS instrument, but heteroconjugates can also be detected by an Orbitrap instrument. Thus, we envisage that this PD-XLMS method will be widely applicable to dissecting the molecular mechanisms of other biologically relevant protein–DNA transactions.

Taken together, our structural, biochemical and biophysical results reveal how Tdp1(Δ148) engages a physiological DNA duplex and removes a 3′-nucleoside to produce a 3′-phosphorylated DNA end. They also suggest a structural mechanism for the lack of enzyme processivity, which is crucial to Tdp1's function in protecting 3′-phosphorylated DNA ends.

Tdp1 removes 3′-DNA lesions from a diverse range of DNA structures, independent of DNA sequence. It cleaves 3′-phosphotyrosine bonds faster from blunt-ended duplexes than single-stranded DNA substrates in vitro, but binds to each substrate with similar affinity[39]. By contrast, nicked or 5′-tailed duplex DNA substrates are cleaved poorly in vitro (with a turnover rate tenfold lower than single-single-stranded substrates). These findings are consistent with removal of Top1 peptides attached at 3′-termini of DSBs, formed after collision of the SSB with a replication fork[4,40], as well as Top1 peptides attached at chromosomal single-strand breaks. In support of this, human Tdp1 interacts directly with DNA ligase IIIα and co-immunoprecipitates with PNK and XRCC1, suggesting that Tdp1 acts within a complex as part of a SSB repair pathway[27]. Interestingly, both *Saccharomyces cerevisiae* Tdp1, and to a lesser extent human Tdp1, are also able to cleave topoisomerase 2-associated 5′-tyrosyl DNA adducts[41,42].

Our new Tdp1(Δ148)–DNA co-crystal structures show how duplex DNA is accommodated by Tdp1(Δ148). This differs from a previous binding model in which the DNA duplex was proposed to bend away from the narrow DNA-binding channel[39]. Our co-crystal structures show that, apart from the surface-exposed β-turn between β15 and β16 (Fig. 7e), Tdp1(Δ148) does not change its shape significantly upon duplex DNA binding. Instead, it sculpts the DNA. Tdp1(Δ148) separates the scissile and complementary DNA strands in the vicinity of the cleavage site, using a hydrophobic loop and intercalation of an aromatic amino acid (F259) to disrupt Watson–Crick base pairing and base-pair stacking. In a similar fashion, other nucleases (e.g., Mre11, Nfi) use alpha helical hydrophobic wedges to locally reshape their DNA substrates and to permit incision[43]. Thus, our results suggest that local melting of the DNA by the Tdp1 hydrophobic loop is required for and permits 3′ DNA-end processing.

The Tdp1(Δ148)–DNA co-crystal structures provide snapshots of the 3′-phosphate product of Tdp1 nucleosidase activity bound in the enzyme's active site. As such they extend our knowledge of the enzyme's mechanism, beyond its tyrosyl-DNA phosphodiesterase activity. Although Tdp1 removes 3′ nucleosides much less efficiently than 3′-phosphotyrosines, phosphoglycolates or CTNAs[20], this activity may nevertheless have a biological role. In *Saccharomyces cerevisiae*, Tdp1 nucleosidase activity was proposed to mediate error prevention during NHEJ[44]. By generating and binding tightly to the 3′-phosphate end, the main function of Tdp1 may be to protect the exposed 3′ terminus while the NHEJ machinery assembles[44]. In support of this proposal, Tdp1 interacts functionally with the NHEJ core factor proteins XLF and Ku70/80, which are recruited to the sites of DSBs[45]. Thus, the nucleosidase activity of Tdp1 may be important for DNA repair.

Tdp1 removes one nucleoside only from a 3′-hydroxyl DNA end, leaving a terminal 3′-phosphate. Our results explain, at a molecular level, why this reaction product is not further hydrolysed by Tdp1. Progressive advancement of the scissile DNA strand (by one nucleotide) into the narrow ssDNA-binding protein channel would be required for the Tdp1 cleavage product to become the next substrate for enzymatic cleavage. This would also require release of the terminal 3′-phosphate from the active site and separation of the −3 base pair. However, by abutting the ds/ssDNA junction, F259 impedes such a movement (Fig. 8a, b): DNA intercalation and π–π stacking of the F259 aromatic ring against the −3A base acts as a wedge, lodging the strand in position and preventing further separation of the duplex. At the other end of the narrow channel, interactions of the 3′-phosphate with the active site residues secure the position of the DNA 3′-end (Fig. 8a). By fixing the strand in this way, processive degradation

of the 3′ end by Tdp1 is prevented, which is essential for Tdp1's biological function in DNA repair.

## Methods

**Tdp1(Δ148) mutation, expression and purification.** N-terminally truncated Tdp1(Δ148), comprising amino acids 149–608, was expressed in *E. coli* strain BL21 (*DE3*) from the plasmid pHN1894s Δ1–148 with a N-terminal hexa-histidine tag[23,46]. Site-directed mutagenesis (Quikchange, Stratagene) of the Tdp1(Δ148) gene generated the substitutions F259A, F259W, F259Y and H263A. Recombinant Tdp1(Δ148) proteins were purified on phosphocellulose P11 resin (Whatman) followed by nickel affinity chromatography. Crude cell extract was loaded onto the P11 resin at 1 ml min$^{-1}$. The buffer contained 30 mM potassium phosphate pH 7 and 100 mM KCl for loading, 400 mM KCl for washing and 800 mM KCl for elution. Fractions containing Tdp1 (Δ148) were dialysed against 0.5 M NaCl, 10 mM imidazole, 20 mM Tris pH 7.9, 0.2 mM PMSF and loaded onto a Hi-Trap Chelating column (1 ml, GE Healthcare). The column was washed with 25 ml of wash buffer (0.5 M NaCl, 30 mM imidazole and 20 mM Tris pH 7.9) and Tdp1 was eluted with 5 ml of buffer containing 0.5 M NaCl, 1 M imidazole and 20 mM Tris pH 7.9. For storage, proteins were exchanged into buffer containing 50 mM Tris pH 7.5, 50 mM KCl, 2 mM DTT and 50% (v/v) glycerol and flash frozen at −80 °C at a final concentration of 55 μM.

**Tdp1(Δ148) cross-linking to DNA oligonucleotides.** Equimolar amounts of Tdp1 (Δ148, H263A) and a DNA oligonucleotide (5IdU(−2), 5IdU(−3) or unmodified control oligonucleotide, with the sequences shown in Table 1) were mixed in buffer containing 10 mM Tris-HCl, pH 7.5 and 100 mM KCl to a final concentration of 5 μM in 500 μl. Prior to cross-linking, samples were incubated at room temperature for 15 min and then on ice for 15 min. Samples were irradiated at 312 nm on ice for 30 min using a UV transilluminator (Vilber-Lourmat T-8M UV-B bulbs).

For SDS-PAGE analysis of cross-linking reactions, 40 μl of each cross-linked sample was mixed with 10 μl 5x Laemmli Buffer (312.5 mM Tris-HCl, pH 6.8, 10% SDS, 50% glycerol, 25% β-mercaptoethanol and 0.005% bromophenol blue) and heated at 95 °C for 3 min. Samples were separated by 8% SDS-PAGE, prepared using the Protogel Kit (EC-890, National Diagnostics). To visualise protein, gels were stained with SimplyBlue™ SafeStain (LC6060, Invitrogen, Life Technologies). For visualisation of DNA on SDS-PAGE, cross-linking reactions were spiked with a small amount of 5′ $^{32}$P-labelled oligonucleotide. Oligonucleotides were 5′ end labelled with $^{32}$P using T4 polynucleotide kinase (PNK, NEB M0201) and [γ-$^{32}$P] ATP (Perkin Elmer NEG002A250UC) in standard 20 μl labelling reactions (10 pmol oligonucleotide, 2 μl PNK buffer, 3 μl [γ-$^{32}$P] ATP, 30 min at 37 °C, followed by inactivation of PNK at 65 °C for 20 min). Gels were transferred onto Whatman 3MM paper, dried under vacuum at 60 °C for 3 h and analysed using a Storm PhosphorImager and the ImageQuant software (GE Healthcare).

**Capture and enrichment of Tdp1(Δ148)–DNA heteroconjugates.** Tdp1 (Δ148)–DNA heteroconjugates were enriched and separated from non-cross-linked oligonucleotides by capture on magnetic Dynabeads® (His-tag Isolation and Pulldown, 10103D, Invitrogen, USA) via the Tdp1(Δ148) N-terminal 6-histidine tag. Dynabeads® (125 μl per sample) were washed three times with 1 ml of Binding Buffer (300 mM NaCl, 50 mM sodium phosphate pH 8.0), before addition of cross-linked sample (500 μl) and incubation at room temperature for 30 min in 750 μl of 2× Binding Buffer and 250 μl H$_2$O. Dynabeads® were washed twice with 1 ml Binding Buffer, then three times with 1 ml 50 mM ammonium bicarbonate.

**On-bead trypsin digestion.** Tdp1(Δ148)–DNA heteroconjugates immobilised on Dynabeads® (370 μl in 50 mM ammonium bicarbonate) were reduced in 500 μM DTT at 50 °C for 15 min, then alkylated in 1 mM iodoacetamide at room temperature for 15 min in the dark. Samples were then digested with 5 μg of sequencing grade trypsin (V5111, Promega) at 37 °C for 20 h. Supernatant containing tryptic peptides and heteroconjugates was removed from the beads and phenylmethanesulfonyl fluoride added to 1 mM to inactivate trypsin.

**Avidin enrichment.** Binding of the trypsin-digested heteroconjugates to monomeric avidin (via the 3′ biotin DNA adduct) enabled their enrichment and the removal of non-cross-linked peptides. To equilibrate the Monomeric Avidin UltraLink Resin (53146, Pierce), 100 μl of slurry (per sample) was washed 3x with 500 μl of 50 mM ammonium bicarbonate in a 0.8 ml microcentrifuge column (89868, Pierce), with centrifugation at 1000 × *g* for 30 s at each step. The monomeric avidin resin was then incubated with trypsin-digested samples in closed centrifuge columns at room temperature for 1 h while mixing, then washed 3x with 500 μl of 50 mM ammonium bicarbonate. Bound heteroconjugates were eluted with 2 × 300 μl of 2 mM D-biotin in 50 mM ammonium bicarbonate, dried in a speed vac and re-suspended in 30 μl H$_2$O.

**Mass spectrometry of Tdp1(Δ148)–DNA heteroconjugates.** Samples were analysed by negative mode C18-HPLC-ESI-FTICR mass spectrometry, as described previously for the analysis of heteroconjugates[36], using reverse-phase high-

performance liquid chromatography (RP-HPLC) and an U3000 HPLC system (Dionex, UK) coupled to the standard electrospray source (Bruker Daltonics) and a SolariX FTICR mass spectrometer equipped with a 12T superconducting magnet (Bruker Daltonics). Acquisition of LC/MS data was controlled by HyStar, version 3.4, build 8 (Bruker Daltonics).

RP-HPLC was performed using a C18 column (Kinetex 2.6 μm C18 100 Å, LC Column 100 × 2.1 mm, 00D-4462-AN, Phenomenex, CA, USA). Heteroconjugates were analysed in negative mode using buffer A (100 mM 1,1,1,3,3,3-hexafluoro-2-propanol (HFIP), 4 mM triethylamine in 5% methanol), and buffer B (100 mM HFIP, 4 mM triethylamine in 100% methanol). The column was run using a flow rate of 200 μl min$^{-1}$. In the first 5 min of the run, a linear ramp from 5% B to 10% B was diverted to waste. From 5 to 20 min, a linear ramp to 30% B was followed by a final ramp to 60% B from 20 to 25 min. The column was washed with 100% B for 2 min and then 5% B for 3 min. For electrospray ionisation, gas pressure was typically ~2.2 psi and spray voltage was ~4.5 kV. For mass spectrometry, ion accumulation times were typically 0.3 s. Ions were trapped using a 6 cm × 10 cm infinity cell. Each individual LC/MS spectrum was the sum of two acquisitions. Transient data size was typically 1 or 2 Mword for each acquisition, and sine-bell multiplication apodization was applied to each transient during FT-MS postprocessing. DNA-containing heteroconjugates were detected by their absorbance at UV 260 nm and corresponding mass spectrum. For each sample, a single fraction was collected from 15 to 17 min, from an identical RP-HPLC run without MS. A control sample, from the cross-linking reaction with unmodified oligonucleotide, was collected at the equivalent time interval. All samples were then dried in a speed vac prior to nuclease P1 digestion.

**Nuclease P1 digestion of heteroconjugates.** Nuclease P1 (250 units, N8630, Sigma-Aldrich) was dissolved in 1 ml of H$_2$O. Heteroconjugate samples were reconstituted in a volume of 20 μl containing 50 mM ammonium acetate, pH 5.2, and 100 nM ZnCl$_2$ and 0.1 units of nuclease P1 and incubated for 30 min at 50 °C.

**Mass spectrometric analysis of digested heteroconjugates.** For offline direct infusion of nuclease P1-digested heteroconjugates, samples were desalted using a ZipTip$_{C18}$ (Millipore) with buffer A (0.1% formic acid) and buffer B (0.1% formic acid/50% acetonitrile). Electrospray ionisation in positive mode was achieved using the Triversa NanoMate (Advion BioSciences) operated in infusion mode coupled to a SolariX FTICR mass spectrometer. Gas pressure was typically ~0.65 psi, and spray voltage was ~1.5 kV. Direct infusion spectra were typically the sum of 100–200 acquisitions. Fragmentation was performed using collision-induced fragmentation in the collision cell. All mass spectra were analysed using Data Analysis software version 4.1 SR1 build 362.7 (Bruker Daltonics). Product ions were calculated and assigned manually.

**Fluorescence-based Tdp1(Δ148) cleavage assays.** Two DNA substrates (named HEI-40 and HEI-50, Table 1) with 5′ 5(6)-carboxyfluorescein (56-FAM) and 3′ Black Hole Quencher (3BHQ) modifications were purchased from Integrated DNA Technologies. Each was dissolved to 50 μM in 10 mM Tris pH 8.0 and 50 mM NaCl. The sequence of the 34 nt HEI-50 is self-complementary, apart from the central 4 nt, and can form a 15 bp double-stranded hairpin structure. To promote hairpin formation and to prevent formation of dimers, HEI-50 was heated to 80 °C for 10 min and snap-cooled rapidly on ice. The sample was analysed on a 12% (w/v) native polyacrylamide gel and the mobility was compared to 16 bp dsDNA and 36 bp dsDNA substrates (Supplementary Fig. 4).

Fluorescence-based Tdp1(Δ148) activity assays were carried out in 96-well black opaque plates (Greiner Bio-One). Each sample (100 μl) contained 50 nM DNA in reaction buffer (100 mM KCl, 10 mM Tris pH 7.5, 1 mM EDTA, 1 mM DTT, 100 μg ml$^{-1}$ BSA). Tdp1(Δ148) was added last to a final concentration of 50 nM or 100 nM in reactions containing the single-stranded or hairpin DNA substrates, respectively. Reactions proceeded for 90 min at 25 °C. The fluorophore was excited at 488 nm and its emission was measured at 523 nm every 30 s on a SpectraMax M5 multi-mode microplate reader (Molecular Devices) using SoftMaxPro software. The data were processed in Microsoft Excel and graphs plotted in GraphPad Prism.

**Preparation of double-stranded DNA substrates for crystallisation.** Two synthetic 12-mer DNA oligonucleotides (Integrated DNA Technologies, Belgium), with the self-complementary sequences 5′ CAC TGC GCA GTG and 5′ CAA TGC GCA TTG (Fig. 5b), were dissolved to 1 mM in buffer comprising 10 mM Tris pH 8.0 and 50 mM NaCl. Each oligonucleotide was annealed to an identical strand by heating to 90 °C for 10 min and then cooling in the heat block to room temperature over 12 h. To confirm annealing, samples were separated on a 20% native polyacrylamide gel, stained with methylene blue and the mobility compared with the single-stranded 12-mer, a single-stranded 16-mer and a double-stranded 14-mer control.

**Preparation of the Tdp1(Δ148):DNA complexes for crystallisation.** Tdp1 (Δ148) complexes with duplex DNA (named −2G or −2T, according to the DNA sequence) were formed by discontinuous diafiltration. Tdp1(Δ148) (100 μl at 55 μM in storage buffer) was diluted into 400 μl of a 22 μM 12-mer duplex DNA in

crystallisation buffer containing 50 mM HEPES pH 7.8, 50 mM NaCl. After incubation on ice for 30 min, the sample was added to a 5000 MCO Vivaspin (500 µl, Sartorius Stedium Biotech), repeatedly concentrated fivefold and then diluted in crystallisation buffer until >99% buffer exchange had been achieved. The final concentration of the complex was 110 µM.

**Crystallisation.** Crystals were grown by sitting-drop vapour-diffusion in MRC maxi 48-well crystallisation plates (Swissci). The drops contained 1.5 µl of Tdp1 (Δ148):DNA complex (110 µM) and 1.5 µl of well solution comprising 0.1 M ammonium sulphate, 0.1 M HEPES pH 7.5 and 25% (w/v) PEG 8000. The crystals were briefly immersed in well solution supplemented with 10% (v/v) glycerol prior to cooling in liquid nitrogen for X-ray diffraction experiments.

**X-ray crystal structure determination and refinement.** X-ray diffraction data were collected on beam line I04 at the Diamond Light Source. The wavelength of the X-rays was 0.9795 Å and 0.92891 Å for the Tdp1 (Δ148) complex with the −2G or the −2T DNA duplex, respectively. Crystals displayed orthorhombic ($P2_12_12$) symmetry and diffracted X-rays to a maximum resolution of 2.04 Å. The X-ray diffraction data were processed with xia2 and the statistics are shown in Table 2. Initial phases were determined by molecular replacement, using the structure of the apo form of Tdp1(Δ148) (PDB ID: 1JY1) as the search model in PHASER. The DNA structure was built manually in Coot. Restrained refinement was performed with Refmac5 and Coot. The refinement statistics are shown in Table 2. For the Tdp1 (Δ148) −2G DNA complex, 96% of the amino acids were in the preferred region of the Ramachandran plot, and the remaining 4% were in the allowed region. For the Tdp1 (Δ148) −2T DNA complex, 97% of the amino acids were in the preferred region of the Ramachandran plot, and the remaining 3% were in the allowed region. All structural diagrams were prepared using PyMOL (http://www.pymol.org/) and Adobe Illustrator.

**Determination of DNA length in crystal by denaturing PAGE.** The 12-mer substrate oligonucleotide −2G 12-mer and five size-marker oligonucleotides, ranging from 7 to 11 nts (with the sequences shown in Table 1), were 5′ end labelled with [32]P in standard 20 µl labelling reactions as described above. In order to generate a 3′-phosphorylated 11-mer from the 12-mer, 1 pmol of [32]P-labelled 12-mer was incubated with 3 µg Tdp1(Δ148) in a total volume of 10 µl at 37 °C for 1 h in Tdp1 reaction buffer (100 mM KCl, 10 mM Tris-HCl pH 7.5, 1 mM EDTA, and 1 mM DTT). The reaction was stopped by addition of 10 µl of formamide-loading dye (96% formamide, 20 mM EDTA, 0.03% xylene cyanol and 0.03% bromophenol blue).

A corresponding 5′ end-labelled 11-mer with a 3′ hydroxyl was generated by first incubating 1 pmol of unlabelled 12-mer with 3 µg Tdp1(Δ148) in a total volume of 10 µl at 37 °C for 1 h, then at 95 °C for 5 min. The resulting 11-mer with a 3′ phosphate was then incubated with PNK, as described above, which adds a 5′ [32]P-label and removes the 3′ phosphate generating a 3′ hydroxyl end. Reactions were stopped with at least an equal volume of formamide-loading dye.

A crystal of the Tdp1(Δ148):DNA complex containing the 12-mer double-stranded DNA oligonucleotide sc12mer_G(-2), grown for ~80 days at 17 °C, was washed twice in crystallisation well solution and then dissolved in 1 µl of water to an estimated DNA oligonucleotide concentration of 0.5 pmol µl$^{-1}$. This DNA was 5′ end labelled in a total volume of 10 µl by incubating with 0.5 µl of PNK and 0.75 µl [γ-32P] ATP at 37 °C for 30 min, and then at 65 °C for 20 min to inactivate PNK. One µl of the labelling reaction containing ~50 fmol oligonucleotide was added to 4 µl of formamide-loading dye and used for analysis.

All samples were denatured for 3 min at 95 °C and 5 µl of each, containing 50 fmol oligonucleotide, were separated on 20% denaturing UREA PAGE as described[47]. Image retrieval was carried out using a PhosphorImager and IMAGEQUANT software (Molecular Dynamics).

**Data availability.** The X-ray data generated during this study are available as public data sets at the RCSB protein databank with accession numbers 5NW9 and 5NWA. The biochemical data and mass spectrometry data are available as public data sets[48,49] at the University of Edinburgh DataShare repository at https://doi.org/10.7488/ds/2237 and https://doi.org/10.7488/ds/2243, respectively. All data are also available from the corresponding authors on request.

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

## Acknowledgements

We thank the beam line staff at Diamond Light Source (beam lines I04 and I04-1) and David Leach, Atlanta Cook and David Finnegan for their comments on the manuscript. The work was funded by a Medical Research Council Career Development Award (G0700257) and a Scottish Universities Life Sciences Alliance (SULSA) grant to H.I. A Wellcome Trust University of Edinburgh Institutional Strategic Support Fund award to J. M.R. supported L.A.A. The BBSRC supported E.R.M. (grant BJ000884 to J.M.R.) and F.J. F. (postgraduate studentship).

## Author contributions

F.F.: Conception and design of PD-XLMS experiments; acquisition, analysis and interpretation of data; and revising the article. E.R.: Acquisition, analysis and interpretation of data. L.A.A.: Acquisition, analysis and interpretation of data. S.B.: Acquisition, analysis and interpretation of data. R.C.: Contributed essential reagents. E.R.M.: Acquisition, analysis and interpretation of X-ray data; and revising the article. C.L.: Design of mass spectrometry experiments; and analysis and interpretation of data. H.I.: Conception and design; analysis and interpretation of data; and drafting and revising the article. J.M.R.: Conception and design; acquisition, analysis and interpretation of data; and writing the manuscript.

## Additional information

**Competing interests:** The authors declare no competing financial interests.

