## [Peer Review File · Nature Communications]

Reviewers' comments:

Reviewer #1 (Remarks to the Author):

Structural basis for DNA 3'-end processing by human Tyrosyl-DNA phosphodiesterase 1

Fiona J. Flett, Emilija Ruksenaite, Shipra Bharati, Elizabeth R. Morris, Roberta Carloni, C. Logan Mackay, Heidrun and Julia M. Richardson

In this manuscript the Authors apply a novel site-specific UV cross-linking technique. Using mass spectrometry, Tdp1 induced DNA damage response was investigated by identification of DNA protein interactions. Based on the cross-linking data, they describe the Tdp1 DNA binding properties, nucleosidase activity and confirm their findings with a new crystal structure of Tdp1(Δ 148) in complex with physiological dsDNA. Here, the conserved phenylalanine site F259 is especially required for efficient DNA processing.

Previous publications of the lab have already established click chemistry generated DNA-peptide heteroconjugates as an MS tool for identification of site-specific DNA-Peptide interactions. Data in this manuscript extend the knowledge on DNA-Protein interactions using state of the art MS techniques. Further, the authors used fluorescence based activity assays to confirm the activity of purified wild-type and mutated Tdp1(Δ 148).

I recommend publication in Nature Communications and request to address the following comments and revision of some parts of the manuscript.

Major Comments:

1. Since 'the DNA substrate of a biological Tdp1 substrate is most likely a nicked double stranded species': is the site-specific UV cross-linking reaction also feasible with a hairpin DNA substrate similar to fluorescence activity assay? By positioning the 5IdU in position -3 of the complementary strand, the DNA processing and the role of the hydrophobic wedge would become clearer. If 5IdU was in position -1 of the scissile strand, would it cross-link to H263 and H493? This could underline the power of the method and increase the confidence of identified cross-link peptides.
2. The Author did not indicate the FDR of peptide identifications or the peptides that are in the database. How many cross-link specific peptides were observed in total? Since there is only one unique tryptic peptide I suggest to use alternative enzymes for protein digestion. This increases

the confidence in mass spectrometry data by confirming the protein sequence – e.g. AspN + GluC.

3. “Tdp1(Δ 148) separates the scissile and complementary DNA strands in the vicinity of the cleavage site, using a hydrophobic loop (containing F259)[...]”: Are neutral mutants F259W or F259Y less active due to lower hydrophobicity, hence, is the phenylalanine specific π stacking essential for disruption of DNA? This could be tested in a second fluorescence activity assay.

Reviewer #2 (Remarks to the Author):

This manuscript by Flett and coworkers addresses the mechanism of a key protein involved in DNA repair, Tyrosyl-DNA phosphodiesterase 1 or Tdp1. Although the biological role of Tdp1 and its structure and catalytic mechanism have been thoroughly analyzed, the manner in which it binds its DNA substrate is poorly understood. This is important because Tdp1 has the remarkable ability to cleave all manner of 3' adducts from the end of DNA while leaving the 3' phosphate group intact and proceeding no further down the DNA chain. This manuscript fully explains this property of Tdp1 using very elegant crystallography and associated biochemical/biophysical methods. Tdp1 has been the focus of many small molecule therapeutics studies to synergistically assist in cancer radiation therapies, and the work is therefore significant for both scientific and medical reasons.

A number of groups have tried unsuccessfully to visualize the biologically relevant Tdp1-DNA complex, and the authors have succeeded by using a palindromic DNA construct that resulted in crosslinking two Tdp1 molecules in the crystal lattice. Although somewhat fortuitous, it was nevertheless a clever strategy that deserved the eventual outcome. The structure answers all of the outstanding questions of Tdp1; it shows how Tdp1 pries open the end of DNA, directs the 3' end into the active site, accommodates the dsDNA, and prevents further ingress of the DNA into the active site locale. The latter is particularly important and is achieved by a strategically located and highly conserved phenylalanine side chain that stacks on the DNA base at the ss-dsDNA junction. The structure is supported by a clever and new crosslinking/mass spec approach that will be applicable to mapping out many other protein-DNA complexes. The description of this new method is an important technical aspect of this paper. The crystallographic interpretation of the electron density did require some assumptions about how Tdp1 processed the bound DNA that have important implications for the mechanism, but these are entirely reasonable and fully supported by some impressive analyses of the DNA species in the crystal. Overall, this is impressive work and includes an author with unique experience of Tdp1, its biochemistry and the important outstanding questions (Heidrun Interthal). I recommend publication with the following minor caveats.

The way in which the manuscript is written, with the crosslinking first and the crystallography second, suggests that the mapping results helped the decision about which DNA construct to use. In fact, the mapping only revealed one interaction, albeit with the all important F259. Does the structure suggest another crosslinking site that could further confirm the structure? The structure does not support an earlier model in which the bound dsDNA is bent, as noted by the authors, but the crystal lattice/dimeric structure may not allow this.

The authors briefly state (or suggest - line 121) that the crosslinking method is only applicable to aromatic amino acids. Is this true and why? This was ideal for F259, but would limit the technique in general. This may also limit further mapping of the Tdp1-DNA complex.

The paper is very well written, of the appropriate length and contains excellent figures.

Reviewer #3 (Remarks to the Author):

Review of Flettt et al., “Structural basis for DNA 3'-end processing...

The authors describe a novel approach to understand protein DNA interactions by “site-specific protein-DNA cross-linking with mass spectrometry”. They apply this technique to understanding the interactions of tyrosyl DNA phosphodiesterase 1 with DNA. Importantly, the observations obtained with this technique are buttressed by structural studies and to some extent, with biochemical studies as well. The work is novel, and the described technique will be of interest to workers in DNA enzymology. The analysis of Tdp1 will be of interest to the DNA repair community, and to workers on topoisomerase targeting anti-cancer agents.

Scientifically, I have very minor issues that the authors need to address. The authors find that the Phe259Ala mutant is defective on both single and double stranded substrates. The authors need to provide a clear explanation for this result.

I am a bit confused by the authors' contention that they describe a way that Tdp1 provides end-protection. I agree that their results provides an additional wrinkle to this point, but I would imagine that the generation of a 3' PO₄ is likely sufficient to generate end-protection, and I think the authors need to discuss this explicitly.

I am not sure I completely agree with the contention in the discussion that in yeast, Tdp1 removed Top1 from double strand breaks, whereas in mammalian cells it acts predominantly on

single strand breaks. To my knowledge, there is no clear evidence from yeast that Tdp1 does not also act on trapped Top1 at single strand breaks. Just because yeast lacks the dedicated machinery found in mammalian cells does not imply a lack of single strand break repair pathways. Similarly, I know of no clear evidence that Tdp1 does not have a role in processing Top1 at double strand breaks.

While the present manuscript deals with processing 3' adducts, the authors need to mention the significant literature that suggests Tdp1 can process 5' adducts as well.

Finally a note about the overall presentation. I think Figure 7, panels A-C are difficult to interpret, and I would suggest the authors rethink their presentation of these figures.

Minor point Page 10 line 256 P^(superscript 35). I imagine this is a typo?

Reviewers' Comments:

Reviewer #1 (Remarks to the Author):

Structural basis for DNA 3'-end processing by human Tyrosyl-DNA phosphodiesterase 1

Fiona J. Flett, Emilija Ruksenaite, Shipra Bharati, Elizabeth R. Morris, Roberta Carloni, C. Logan Mackay, Heidrun and Julia M. Richardson

In this manuscript the Authors apply a novel site-specific UV cross-linking technique. Using mass spectrometry, Tdp1 induced DNA damage response was investigated by identification of DNA protein interactions. Based on the cross-linking data, they describe the Tdp1 DNA binding properties, nucleosidase activity and confirm their findings with a new crystal structure of Tdp1(Δ 148) in complex with physiological dsDNA. Here, the conserved phenylalanine site F259 is especially required for efficient DNA processing. Previous publications of the lab have already established click chemistry generated DNA-peptide heteroconjugates as an MS tool for identification of site-specific DNA-Peptide interactions. Data in this manuscript extend the knowledge on DNA-Protein interactions using state of the art MS techniques. Further, the authors used fluorescence based activity assays to confirm the activity of purified wild-type and mutated Tdp1(Δ 148). I recommend publication in Nature Communications and request to address the following comments and revision of some parts of the manuscript.

Major Comments:

1. Since 'the DNA substrate of a biological Tdp1 substrate is most likely a nicked double stranded species': is the site-specific UV cross-linking reaction also feasible with a hairpin DNA substrate similar to fluorescence activity assay? By positioning the 5IdU in position -3 of the complementary strand, the DNA processing and the role of the hydrophobic wedge would become clearer. If 5IdU was in position -1 of the scissile strand, would it cross-link to H263 and H493? This could underline the power of the method and increase the confidence of identified cross-link peptides.

Yes, this is a good suggestion. In principle, a hairpin substrate similar to the one used for the fluorescence activity assays with 5IdU in various positions should be a good substrate for cross-linking experiments. Alas, our preliminary cross-linking experiments with radiolabelled hairpin oligonucleotides (containing a biotin moiety as the 3' adduct and modified with 5IdU in the -2 position on either the scissile or the complementary strand) did not result in cross-linked material that could be detected by SDS-PAGE and phosphorimaging. This could have been due to the particular oligonucleotides that we used, as they were somewhat heterogeneous. Alternatively, the local environment in the protein-DNA complex may have prevented cross-linking. Unfortunately, in the end only the presence of a cross-link can be meaningfully interpreted, not its absence.

The -3 base on the complementary strand is $>8 \text{ \AA}$ away from F259, therefore we would not expect it to cross-link to F259 and there is no other aromatic residue nearby in our structures.

We have not tried 5IdU in the -1 position of the scissile strand, but we would predict that this base is more likely to cross-link to W590 than H493, as it is closer ($\sim 3.4 \text{ \AA}$) than H493, which may be too far away ($>8 \text{ \AA}$). The other active site histidine - H263 - is mutated to alanine in our cross-linking experiments, to inactivate the Tdp1 enzyme. Thus, it is not available for cross-linking to 5IdU.

2. The Author did not indicate the FDR of peptide identifications or the peptides that are in the database. How many cross-link specific peptides were observed in total? Since there is only one unique tryptic peptide I suggest to use alternative enzymes for protein digestion. This increases the confidence in mass spectrometry data by confirming the protein sequence – e.g. AspN + GluC.

Because our data analysis was performed manually with a small dataset, we cannot determine an FDR for these experiments. However, we are confident that we have identified the correct cross-linked peptides in these experiments:

In order to identify potential cross-linked Tdp1 peptide-DNA oligonucleotide heteroconjugate species, we collected the LC fractions that showed absorbance at 260 nm, indicating the presence of DNA (see Fig. 3C and D). Using negative mode MS we then manually identified four potential heteroconjugate species, species with a mass greater than the mass of the oligonucleotide alone. Three of those species were of such low abundance that we were unable to unambiguously identify the monoisotopic peaks and determine their exact mass. For the fourth, and by far most abundant, species, we could determine the exact mass and from there calculate the mass of the potentially cross-linked peptide. This mass was searched against an *in silico* generated database of tryptic Tdp1 peptides (which allowed for up to two missed cleavages) and fit with the L₂₅₅DIAFGTHATK₂₆₅ peptide that was later confirmed.

We have amended the text of the manuscript on page 7 to indicate that more than one potential heteroconjugate was identified using LC/MS to:

“The corresponding masses of the most abundant potential heteroconjugates in this time interval were 7927.11 Da and 7912.11 Da in the -3 and -2 cross-linked samples, respectively (Figure 3D).”

We appreciate the reviewer’s suggestion that using alternative proteases in addition to trypsin could increase the confidence in the correct identification of a single cross-linked species. We would argue though, that with the data presented in this manuscript, we can be very confident that we have correctly identified the main peptide that specifically

cross-links with the 5IdU in the oligonucleotide, without having to expand our repertoire of proteases.

- We have identified two different heteroconjugate species, with different masses (due to different oligo sequences), both containing the L₂₅₅DIAFGTHATK₂₆₅ peptide cross-linked to -2 and -3 modified oligonucleotide, respectively.
- We fragmented (CID) and sequenced these two heteroconjugates after nuclease digestion and obtained full fragment coverage of the peptide-dUMP heteroconjugates with all y and z ions detected, including the F259 modified by the link to dUMP, for both (Fig. 3F).
- We have added the following sentences to page 8 of the manuscript to show that we could detect additional ions for each heteroconjugate and were able to fragment and sequence them as well :

“Further confirming this cross-link, in the MS spectrum shown in Figure 3E, we also detected and fragmented the [M+3H]³⁺ ion (m/z 493.89) of this cross-link.”

- Furthermore, we observed and sequenced additional species that contained a dinucleotide cross-linked to the F259 containing peptide and have added the following sentence to page 8.

“Additionally we detected the [M+2H]²⁺ and [M+3H]³⁺ ions (m/z 892.36 and m/z 595.24, respectively) of a heteroconjugate between F259 and a nucleotide dimer of dUMP and the adjacent dTMP, following incomplete digestion of the DNA.”

- Consistent with the identification of F259 in the L₂₅₅DIAFGTHATK₂₆₅ peptide as the cross-linking amino acid residue, we could not detect a cross-link between the mutant Tdp1 F259A protein and the -2 or-3 oligonucleotides (data not shown).

3. “Tdp1(Δ148) separates the scissile and complementary DNA strands in the vicinity of the cleavage site, using a hydrophobic loop (containing F259)[...]”: Are neutral mutants F259W or F259Y less active due to lower hydrophobicity, hence, is the phenylalanine specific π stacking essential for disruption of DNA? This could be tested in a second fluorescence activity assay.

As suggested we performed both fluorescence activity assays with two additional Tdp1(Δ148) mutants: F259W and F259Y. These new data are shown in Figure 4. The F259W mutant has similar cleavage activity to wild-type (WT) enzyme, whereas the F259Y mutant has slightly enhanced activity. This contrasts with the F259A mutant, which displays impaired activity. We conclude that the aromatic nature of F259 and its ability to

π -stack with the -3 base is important for activity, but that the π -stacking is not phenylalanine-specific. Thus, the aromaticity of F259 is likely more important for enzyme function than its hydrophobicity. We have added new text on page 9, which relates to the fluorescence assay results, and on page 14, where the structural role of F259 is discussed.

Reviewer #2 (Remarks to the Author):

This manuscript by Flett and coworkers addresses the mechanism of a key protein involved in DNA repair, Tyrosyl-DNA phosphodiesterase 1 or Tdp1. Although the biological role of Tdp1 and its structure and catalytic mechanism have been thoroughly analyzed, the manner in which it binds its DNA substrate is poorly understood. This is important because Tdp1 has the remarkable ability to cleave all manner of 3' adducts from the end of DNA while leaving the 3' phosphate group intact and proceeding no further down the DNA chain. This manuscript fully explains this property of Tdp1 using very elegant crystallography and associated biochemical/biophysical methods. Tdp1 has been the focus of many small molecule therapeutics studies to synergistically assist in cancer radiation therapies, and the work is therefore significant for both scientific and medical reasons.

A number of groups have tried unsuccessfully to visualize the biologically relevant Tdp1-DNA complex, and the authors have succeeded by using a palindromic DNA construct that resulted in crosslinking two Tdp1 molecules in the crystal lattice. Although somewhat fortuitous, it was nevertheless a clever strategy that deserved the eventual outcome. The structure answers all of the outstanding questions of Tdp1; it shows how Tdp1 pries open the end of DNA, directs the 3' end into the active site, accommodates the dsDNA, and prevents further ingress of the DNA into the active site locale. The latter is particularly important and is achieved by a strategically located and highly conserved phenylalanine side chain that stacks on the DNA base at the ss-dsDNA junction. The structure is supported by a clever and new crosslinking/mass spec approach that will be applicable to mapping out many other protein-DNA complexes. The description of this new method is an important technical aspect of this paper. The crystallographic interpretation of the electron density did require some assumptions about how Tdp1 processed the bound DNA that have important implications for the mechanism, but these are entirely reasonable and fully supported by some impressive analyses of the DNA species in the crystal. Overall, this is impressive work and includes an author with unique experience of Tdp1, its biochemistry and the important outstanding questions (Heidrun Interthal). I recommend publication with the following minor caveats.

The way in which the manuscript is written, with the crosslinking first and the crystallography second, suggests that the mapping results helped the decision about

which DNA construct to use. In fact, the mapping only revealed one interaction, albeit with the all important F259. Does the structure suggest another crosslinking site that could further confirm the structure?

Based on the structure one might predict that the -1 and -2 bases of the scissile strand could cross-link to W590 and Y204, respectively, as they are about 3.4 Å and 2.9 Å away, respectively. With our analysis, using the -2 modified oligonucleotide we have not detected a cross-link between the -2 base and Y204. This could be because the -2 base and the aromatic ring of the tyrosine residue are in a linear arrangement, rather than in the π - π stacking orientation which is most efficient for cross-linking.

The structure does not support an earlier model in which the bound dsDNA is bent, as noted by the authors, but the crystal lattice/dimeric structure may not allow this. The authors briefly state (or suggest - line 121) that the crosslinking method is only applicable to aromatic amino acids. Is this true and why?

We apologize for neglecting to include an appropriate references for UV cross-linking to 5IdU in our manuscript. We have now added two references (29 and 30) on page 6 which explain that 5IdU has been observed mainly cross-linked to aromatic amino acids, the only exception being methionine. The exact mechanism of UV-induced cross-linking is not always known, but observations suggest that π - π interactions are important for achieving high cross-linking efficiency. Thus it seems reasonable to expect 5IdU to cross-link to aromatic residues, but to keep in mind that it may cross-link to other amino acids as well.

This was ideal for F259, but would limit the technique in general. This may also limit further mapping of the Tdp1-DNA complex.

The reviewer is correct, the choice of photoactivatable nucleotide invariably comes with limitations. When deciding which modified nucleotide to use one has to take into account a number of variables, such as cross-linking efficiency, selectivity (range of amino acid residues to cross-link to), size of the substitution (that may distort DNA), excitation wavelength (to minimize photo-damage to DNA and proteins), etc. 5IdU proved to be a good choice for Tdp1 as there are a number of aromatic residues accessible on the surface of the protein. There are alternative photoactivatable nucleotides available which could be used in a similar manner; such as 5-bromouridine (which has been cross-linked to alanine and lysine in addition to aromatic residues) or thio- or azido-substituted nucleobases.

Reviewer #3 (Remarks to the Author):

Review of Flettt et al., "Structural basis for DNA 3'-end processing..."

The authors describe a novel approach to understand protein DNA interactions by "site-specific protein-DNA cross-linking with mass spectrometry". They apply this technique to understanding the interactions of tyrosyl DNA phosphodiesterase 1 with DNA.

Importantly, the observations obtained with this technique are buttressed by structural studies and to some extent, with biochemical studies as well. The work is novel, and the described technique will be of interest to workers in DNA enzymology. The analysis of Tdp1 will be of interest to the DNA repair community, and to workers on topoisomerase targeting anti-cancer agents.

Scientifically, I have very minor issues that the authors need to address. The authors find that the Phe259Ala mutant is defective on both single and double stranded substrates. The authors need to provide a clear explanation for this result.

The new data added to Figure 4, on the activity of two 'neutral' mutants F259W and F259Y (requested by reviewer 1) is also relevant to this issue. We have expanded the last paragraph of the results section (on page 14), which draws together the crystal structure and biochemical data, to address this point. The paragraph now reads as below, with the new text highlighted in red:

"Our crystal structures support the results of the in solution cross-linking mass spectrometry experiments, which identified short range site-specific interactions between F259 and modified nucleobases at the -2 and -3 positions (Figure 3). The role of F259 is also consistent with the results of our *in vitro* biochemical assays, which show that an aromatic amino acid side-chain at position 259 is required for efficient DNA 3'-end processing (Figure 4) of both single- and double-stranded substrates. We envisage that, for cleavage of the single-stranded DNA substrate, the key role of the aromatic amino acid at 259 is to intercalate between nucleobases -2 and -3 and to position to scissile strand in the active site for cleavage. However, cleavage of a double-stranded substrate involves additional reshaping of the substrate, including separation of the two strands of the DNA duplex by the hydrophobic loop and stabilisation of the melted conformation by π - π stacking with the aromatic ring of F259. "

I am a bit confused by the authors' contention that they describe a way that Tdp1 provides end-protection. I agree that their results provides an additional wrinkle to this point, but I would imagine that the generation of a 3' PO4 is likely sufficient to generate end-protection, and I think the authors need to discuss this explicitly.

Thank you for pointing this out. We think that the confusion arises because of a lack of distinction in our manuscript between end-protection from Tdp1 itself (i.e. lack of processivity) and end-protection from other nucleases, which will be conferred simply by the generation of the 3'-phosphate. We meant the former, and so we have removed the final sentence of the discussion and changed the previous sentence to read "By fixing the strand in this way, processive degradation of the 3' end by Tdp1 is prevented". We have also removed the same point from the abstract and replaced it with the sentence "Our results explain why Tdp1 cleavage is non-processive and provide a molecular basis for DNA 3'-end processing by Tdp1"

I am not sure I completely agree with the contention in the discussion that in yeast, Tdp1 removed Top1 from double strand breaks, whereas in mammalian cells it acts predominantly on single strand breaks.

To my knowledge, there is no clear evidence from yeast that Tdp1 does not also act on trapped Top1 at single strand breaks. Just because yeast lacks the dedicated machinery found in mammalian cells does not imply a lack of single strand break repair pathways. Similarly, I know of no clear evidence that Tdp1 does not have a role in processing Top1 at double strand breaks.

We agree with the reviewer and have modified the discussion, in the last paragraph on page 15, to address this point. The relevant sentence now reads: "These findings are consistent with removal of Top1 peptides attached at 3'-termini of DSBs, formed after collision of the SSB with a replication fork as well as Top1 peptides attached at chromosomal single-strand breaks."

While the present manuscript deals with processing 3' adducts, the authors need to mention the significant literature that suggests Tdp1 can process 5' adducts as well.

We have added this important point, along with references (41 and 42) to two papers in the literature, to the discussion on page 16. The sentence reads: "Interestingly, both *Saccharomyces cerevisiae* Tdp1, and to a lesser extent human Tdp1, are also able to cleave topoisomerase 2 associated 5'-tyrosyl DNA adducts^{41, 42}."

Finally a note about the overall presentation. I think Figure 7, panels A-C are difficult to interpret, and I would suggest the authors rethink their presentation of these figures.

Figure 7 panels A-C have been altered in the following ways:

- a) The titles above each structure have been amended to indicate which complexes contain duplex DNA and which structure is the transition-state complex with single-stranded DNA.

- b) The thickness of the sticks representing the DNA strands has been increased.
- c) We have numbered the bases on each strand of DNA on each of the panels A-C.
- d) The sequence of the DNA shown below each structure now indicates only the bases that were visible in that structure. Previously this was represented with shading, which has been removed. This representation now ties in with and follows on from the DNA schematic in Figure 5B.

We hope that these changes make it easier for the reader to follow the description of the relevant results on pages 11 and 12 of the manuscript.

Minor point Page 10 line 256 P(superscript 35). I imagine this is a typo?

This superscript 35 referred to reference 35, so to avoid any confusion we have spelled out phosphate instead (and the reference is now number 37).

We have also added a new supplementary figure 4 showing a stereo image of the electron density map in the vicinity of the active site. Supplementary Figures 5 and 6 have been re-numbered accordingly.

We have added a data availability statement at the end of the materials and methods, which includes doi links to an University of Edinburgh public repository and the Protein Data Bank codes for the new structures reported.

Reviewers' Comments:

Reviewer #1 (Remarks to the Author):

The Authors adequately addressed all comments and performed the suggested experiments. The additional mass spec. information provided, is in my opinion sufficient for the drawn conclusions. The inclusion of neutral mutants in the fluorescence activity assay nicely adds to the interpretation of the mechanisms of Tdp1. I recommend publication of the paper.

Reviewer #2 (Remarks to the Author):

The revised manuscript by Flett and coworkers on the structure and mechanism of tyrosyl-DNA phosphodiesterase 1 has responded very well to my comments and those of the other reviewers. The comments were relatively minor and all of the reviewers recommended publication with some revisions. These have all been addressed in my opinion, and the paper is now ready for publication. I have no further issues with the manuscript that describes excellent and important studies on this key DNA repair enzyme.

Reviewer #3 (Remarks to the Author):

Review of Flett et al., "Structural basis for DNA 3'-end processing..."

The authors describe a novel approach to understand protein DNA interactions by "site-specific protein-DNA cross-linking with mass spectrometry". They apply this technique to understanding the interactions of tyrosyl DNA phosphodiesterase 1 with DNA. Importantly, the observations obtained with this technique are buttressed by structural studies and to some extent, with biochemical studies as well. The work is novel, and the described technique will be of interest to workers in DNA enzymology. The analysis of Tdp1 will be of interest to the DNA repair community, and to workers on topoisomerase targeting anti-cancer agents.

In the previous review, I raised several minor concerns. The authors have responded very well to all of my concerns.